# DO LANGUAGE MODELS TRUST THEIR OWN JUSTIFICATIONS? A STUDY ON FUNCTIONAL CONSISTENCY

## ABSTRACT

Large Language Models (LLMs) have been widely adopted in text classification tasks, where they not only output class predictions but also generate explanations that highlight the tokens deemed most relevant for reaching the predicted label. Yet it remains unclear whether these highlighted elements faithfully reflect the underlying decision process of the model. While much of the literature evaluates the textual plausibility of such explanations, few studies assess their functional consistency with the model's actual behavior. In this work, we propose an experimental framework based on the principle of auto-consistency: if a model identifies certain tokens as decisive, then isolating, removing, or semantically inverting them should produce systematic and interpretable changes in its predictions. We operationalize this evaluation through sufficiency, comprehensiveness, and counterfactuality metrics, and conduct experiments on IMDB and Steam reviews across both closed-source (GPT-4o) and open-source LLMs (Gemma3, Granite8B, DeepSeek). Results show that GPT-4o follows the expected progression across all metrics, Gemma3 and Granite8B maintain coherence under sufficiency but lose consistency under more demanding interventions, while DeepSeek variants display structural deviations, either failing to preserve sufficiency or overreacting under comprehensiveness and counterfactuality. These findings show that explanation reliability varies across LLM families and scales, with smaller models displaying contradictions and larger ones exhibiting over-sensitivity. By combining sufficiency, comprehensiveness, and counterfactuality, our approach provides a systematic methodology for assessing the functional consistency of LLM self-explanations.

## 1 INTRODUCTION

*Large Language Models* (LLMs), built upon the *Transformer* architecture (Vaswani et al., 2017), have demonstrated remarkable abilities in generalization and task solving through textual instructions, even in scenarios where no explicit training examples are available (*zero-shot* or *few-shot learning*) (Brown et al., 2020). Among their emergent capabilities, one of the most prominent is the generation of explanations that justify predictions, often structured as reasoning chains, commonly referred to as *chain-of-thought* (CoT) (Wei et al., 2023).

While such explanations are frequently plausible, their faithfulness to the actual inferential processes of the model remains an open question. For instance, Turpin et al. (2023) show that CoT explanations can be manipulated through biases in the prompts, thereby justifying incorrect predictions with arguments that appear convincing but are not aligned with the true decision factors. These findings raise concerns about excessive reliance on LLM-produced explanations and highlight the need for more objective approaches to evaluate the transparency and robustness of these models.

In this work, we propose an investigation that leverages *feature importance* (Barbieri et al., 2024) within the task of classification with LLMs, aiming to assess the extent to which models remain coherent with the justifications they provide for their own predictive decisions. This evaluation adapts the definition of self-consistency proposed by Chen et al. (2023), understood as the requirement that model outputs remain logically non-contradictory, to what we term auto-consistency. In our setting,

this notion captures whether the tokens[1] highlighted as decisive in a model's explanation play a functionally coherent role when subjected to systematic interventions.

We hypothesize that when certain tokens are highlighted as influential for a prediction, the model should exhibit systematic variations in the assigned score when these elements are removed, isolated, or semantically modified. Thus, explanations are not evaluated in terms of textual plausibility or rhetorical adequacy, but rather in terms of their functional coherence with the model's own predictive behavior.

To quantify this coherence, we adopt the metrics of Comprehensiveness, Sufficiency, and Counterfactuality (Molnar, 2020). The first two follow the definitions of DeYoung et al. (2019), but are here adapted to a scalar review variable. Unlike probability-based formulations, we use a review score in the range $[1, 10]$, predicted by the LLM, which reflects how favorably the text evaluates the target item.[2] Under this formulation, Comprehensiveness captures the variation in the score when the highlighted tokens are removed, Sufficiency evaluates the case in which only these tokens are preserved, and Counterfactuality measures the impact of substituting them with semantically opposite terms.

With this adaptation, we investigate whether the model behaves consistently with its own claims of importance, that is, whether it maintains functional coherence under local interventions on the tokens it designates as relevant. This study therefore focuses on the structural fidelity of explanations provided by LLMs, adopting a quantitative approach to the evaluation of interpretability in generative models. The empirical investigation is conducted in the context of sentiment classification, and the results are discussed in detail in the following sections.

## 2 RELATED WORK

The recent expansion of LLMs in tasks of feature selection has generated a variety of proposals based on different assumptions about their ability to interpret textual descriptions and establish conceptual relationships between variables and predictive tasks. Some of these works, such as LM-Priors (Choi et al., 2022), propose the use of LLMs as a source of prior knowledge about the task, preceding the actual selection of attributes. The central idea is to instruct the model to judge, based solely on descriptions, whether a given attribute should be considered relevant, as also demonstrated by Brown et al. (2020) using binary prompts such as "Yes" or "No."

Other approaches, such as those described in Li & Xiu (2025), operate in a hybrid paradigm, where LLMs are instantiated as mechanisms for generating feature importance but guided by instructions to apply traditional algorithms, such as random forests or sequential selection. This line of work seeks to exploit the semantic expressiveness of LLMs while retaining the statistical consistency of classical techniques.

More sophisticated directions can be found in Yang et al. (2024), which incorporate LLMs into iterative optimization cycles in medical settings, using multiple prompts to refine attribute selection through successive feedback. Complementarily, Han et al. (2024) treat LLMs as feature engineers, using them to generate meta-features that subsequently feed conventional models in order to enhance downstream predictive tasks.

In parallel, a growing body of work has examined the ability of LLMs to explain their own decisions through self-explanations or rationalizations. Studies such as Madsen et al. (2024) and Huang et al. (2023) assess the degree to which model-generated explanations are consistent or useful for interpretability. However, evidence from Sarkar (2024) indicates that such explanations often fail to reflect the underlying inferential process, exposing a gap between textual narratives and the actual decision mechanisms.

In the same critical vein, Turpin et al. (2023) demonstrate that chain-of-thought reasoning can generate explanations that, while linguistically coherent, systematically diverge from the actual inferential processes. Their analysis reveals that models may introduce spurious but plausible reasoning steps,

---

[1]In this paper, the term "token" is used broadly to denote an influential text span (a word or phrase) returned in the influential terms field and extracted verbatim from the input sentence. All interventions operate on these spans, consistent with the prompt specification ("words or phrases"; see Appendix A).

[2]Where 1 indicates an extremely negative evaluation and 10 an extremely positive evaluation.

thereby producing post-hoc narratives that obscure rather than reveal the underlying decision factors. Likewise, Chen et al. (2025) show that even in controlled settings with explicit cues embedded in the input, state-of-the-art reasoning models frequently exploit these cues in their predictions without incorporating them into their explanations. This discrepancy highlights a fundamental misalignment: predictive behavior is often guided by information that remains absent from the articulated justification. Collectively, these findings indicate that the textual plausibility of self-explanations is insufficient as a criterion of evaluation, since models can produce persuasive accounts that mask structural inconsistencies between explanation and decision-making.

Overall, existing work converges on exploiting the semantic reasoning capabilities of LLMs to inform or guide the selection of important features, differing mainly in their reliance on data, integration within the pipeline, and computational complexity. Yet few studies focus on the critical assessment of a model's internal coherence with respect to the explanations it produces.

In this work, we aim to fill this gap by employing feature importance not as the central object of analysis but as a methodological mechanism for evaluating model auto-consistency. Specifically, we apply the metrics of Comprehensiveness, Sufficiency, and Counterfactuality to the tokens highlighted as relevant, thereby assessing whether these elements maintain functional correspondence with the model's observed predictive behavior.

## 3 METHODOLOGY

This section outlines the methodology adopted to evaluate the functional consistency of explanations provided by LLMs in the context of classification tasks. We begin by describing the datasets employed and the sampling criteria used for the selection of sentences analyzed in the experiments. Next, we present the structure of the prompts designed to instruct the model to perform sentiment classification and to identify the tokens considered most relevant for its predictions. Finally, we introduce the metrics used to quantify the fidelity of the explanations, with emphasis on the formal definitions of Comprehensiveness, Sufficiency, and Counterfactuality, which allow us to measure the extent to which the presence, absence, or semantic inversion of explanatory elements affects the model's behavior.

### 3.1 ANALYSIS PIPELINE

The experiments were conducted using two datasets widely employed in sentiment analysis tasks. The first is the *Stanford Large Movie Review Dataset* (IMDB) (Maas et al., 2011), which consists of English-language movie reviews annotated with binary sentiment polarity (positive or negative), used here in the version available on the *HuggingFace* platform. [3] The second dataset comprises reviews from the Steam platform (Pandey & Joshi, 2022), which collects user evaluations of PC games and enables the analysis of aspects such as player satisfaction and dissatisfaction, genre popularity, and sentiment shifts over time.

For the construction of the experimental corpus, a stratified sampling of $2,000$ sentences was performed, ensuring proportionality between sentiment classes and thereby minimizing potential distributional biases. This strategy was adopted to increase representativeness and robustness in the subsequent analyses.

Model interactions were carried out through different APIs, covering recent LLM architectures: *GPT-4o mini* (OpenAI et al., 2024), *Gemma3:4B* (Team et al., 2025), *DeepSeek-R1:1.5B* and *DeepSeek-R1:14B* (DeepSeek-AI et al., 2025)[4] and *Granite3.3:8B* (Mishra et al., 2024). The selection aimed to ensure diversity along two main axes: (i) proprietary versus open-source access, and (ii) providers from distinct research and industrial backgrounds. The additional intra-family comparison was conducted exclusively within the DeepSeek models, given their availability and relevance for methodological contrast.

---

[3]https://huggingface.co/datasets/stanfordnlp/imdb

[4]We additionally include *DeepSeek-R1:14b* to enable an intra-family comparison across parameter scales. This choice was motivated by methodological considerations, since evaluating two models from the same provider with different capacities offers a controlled setting to assess consistency.

For each sentence, the model was instructed to classify the sentiment, assign a **review score** in the range $[1, 10]$, and identify the tokens most relevant for the prediction. This score represents the intensity of the judgment produced by the model and constitutes the basis for the sufficiency, comprehensiveness, and counterfactual metrics discussed in subsequent sections. Figure 1 illustrates this input–output pipeline.

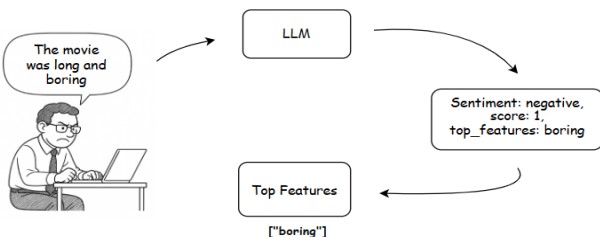

Figure 1: Sentence Analysis Workflow

## 3.2 PROMPT FORMALIZATION

Adapting the formulation in Jeong et al. (2025), let $\mathcal{M}$ denote a pre-trained LLM. The input to $\mathcal{M}$ is a prompt $P^{\text{LLM}}$ composed of three main elements:

- a description of the dataset ($Des$);
- few-shot examples ($Ex$), selected to illustrate the target task;
- the task context ($C$), specifying the objective of identifying the tokens most relevant to the sentiment prediction.

These components are combined to form the complete prompt:

$$P^{\text{LLM}} = \text{prompt}(Des, Ex, C). \tag{1}$$

Let $\mathcal{X} = \{x_1, x_2, \ldots, x_m\}$ be a collection of sentences, where each $x_i = \{w_1^{(i)}, w_2^{(i)}, \ldots, w_{n_i}^{(i)}\}$ consists of $n_i$ tokens. For each $x_i$, we provide $P^{\text{LLM}}$ to the model $\mathcal{M}$ together with the input sentence. The model then returns a subset $\mathcal{T}(x_i) \subseteq x_i$ containing the tokens identified as most relevant to justify its prediction on $x_i$:

$$\mathcal{T}(x_i) = \mathcal{M}(x_i; P^{\text{LLM}}), \qquad i = 1, 2, \ldots, m. \tag{2}$$

In summary, each subset $\mathcal{T}(x_i)$ corresponds to the tokens within $x_i$ that the LLM, when prompted with $P^{\text{LLM}}$, identifies as essential for supporting its sentiment classification. These subsets serve as the basis for the interventions defined in our evaluation metrics, with concrete examples of their application provided in the Appendix A.

## 3.3 METRICS FOR AUTO-CONSISTENCY EVALUATION

The evaluation of LLM explanations is grounded in the principle of **auto-consistency**, defined as the requirement that a model's predictive behavior remains coherent with the set of tokens it designates as influential for its decisions. Formally, interventions applied to the highlighted subset $\mathcal{T} \subseteq x_i$ should induce variations in the assigned review score that are systematic and interpretable, rather than incidental.

This principle is operationalized through three complementary metrics. *Sufficiency* evaluates whether the tokens spans in $\mathcal{T}$, when considered in isolation, are capable of sustaining the original prediction. *Comprehensiveness* quantifies the effect of removing $\mathcal{T}$ from the input, capturing the extent to which the prediction deteriorates in their absence. *Counterfactuality* assesses the impact of replacing $\mathcal{T}$ with semantically opposite terms, thereby determining whether the inversion of

highlighted tokens substantially alters the decision. Together, these metrics provide a formal framework for examining whether the importance attributed by the model to specific tokens is faithfully reflected in its predictive behavior. Here's a specific example from the Steam dataset of sentence modification:

> **Original Sentence:** "Pay to win. Buggy. Doesn't deploy Hero's to lanes, which is kinda the point of the whole game."
>
> **Sufficiency:** "Pay to win. Buggy."
>
> **Comprehensiveness:** "Doesn't deploy Hero's to lanes, which is kinda the point of the whole game."
>
> **Counterfactual Replacement:** "Not pay to win. Stable. Doesn't deploy Hero's to lanes, which is kinda the point of the whole game."

### 3.3.1 COMPREHENSIVENESS

Let $\mathcal{M}$ be a pre-trained LLM. Consider an input sequence $x_i = \{w_1, w_2, \ldots, w_n\}$ consisting of $n$ tokens. Let $\mathcal{T} \subseteq x_i$ denote the subset of tokens highlighted by $\mathcal{M}$, when processing $x_i$ under a given prompt, as the most relevant for explaining its decision.

We denote by $R^{\mathcal{M}}(x)$ the review score assigned by model $\mathcal{M}$ to input $x$, represented as a scalar value in the range $[1, 10]$. This review score reflects the intensity with which the textual content is perceived as favorable to the evaluated item.

We define the *comprehensiveness* metric as:

$$\text{Comprehensiveness}(x_i, \mathcal{T}) = R^{\mathcal{M}}(x_i) - R^{\mathcal{M}}(x_i \setminus \mathcal{T}) \tag{3}$$

where $x_i \setminus \mathcal{T}$ denotes the input sequence with all tokens in $\mathcal{T}$ removed, and both evaluations are performed by the same model $\mathcal{M}$ on the respective texts. [5]

### 3.3.2 SUFFICIENCY

Within the previously established framework, we define the *sufficiency* metric considering the same model $\mathcal{M}$, the input sequence $x_i$, and the subset of tokens $\mathcal{T} \subseteq x_i$ highlighted by the LLM as most relevant to its decision.

Sufficiency is given by:

$$\text{Sufficiency}(x_i, \mathcal{T}) = R^{\mathcal{M}}(x_i) - R^{\mathcal{M}}(\mathcal{T}) \tag{4}$$

where $R^{\mathcal{M}}(\mathcal{T})$ corresponds to the review score resulting from using only the subset $\mathcal{T}$ as input. Large differences between $R^{\mathcal{M}}(x_i)$ and $R^{\mathcal{M}}(\mathcal{T})$ indicate that the highlighted tokens, when considered in isolation, are not sufficient to sustain the original decision, thereby suggesting explanatory insufficiency.

### 3.3.3 COUNTERFACTUALITY

Within the previously established framework, we define the *counterfactuality* metric considering the same model $\mathcal{M}$, the input sequence $x_i$, and the subset of tokens $\mathcal{T} \subseteq x_i$ highlighted by the LLM as most relevant to its decision.

Counterfactuality is defined as:

$$\text{Counterfactuality}(x_i, \mathcal{T}) = R^{\mathcal{M}}(x_i) - R^{\mathcal{M}}(x_i[\mathcal{T} \leftarrow \neg\mathcal{T}]) \tag{5}$$

---

[5]To ensure a directional interpretation consistent with the predicted label, we adopt a symmetric transformation of the review differences ($\Delta R$): when the sentence is labeled as negative, we compute $\Delta R = R_{\text{modified}} - R_{\text{original}}$; otherwise, we compute $\Delta R = R_{\text{original}} - R_{\text{modified}}$. In this way, positive values of $\Delta R$ indicate a reduction in the evaluation toward the negative pole—i.e., an adverse effect of the intervention on the model's perception, regardless of the original label.

where $x_i[\mathcal{T} \leftarrow \neg\mathcal{T}]$ denotes the modified input sequence obtained by replacing the tokens in $\mathcal{T}$ with their semantic opposites.

The construction of $\neg\mathcal{T}$ follows the methodology of semantic editing (Wang et al., 2024): whenever possible, highlighted tokens are replaced by their antonyms retrieved from the lexical database *WordNet* (Fellbaum, 2010). In cases where no suitable antonym exists, we adopt a syntactic negation heuristic by prefixing the token with the operator "not". This procedure ensures a minimally plausible counterfactual intervention, effectively inverting the semantic polarity of the tokens identified as explanatory by the model. High values of Counterfactuality$(x_i, \mathcal{T})$ indicate that the highlighted tokens play a decisive functional role, as their semantic inversion substantially alters the original decision.

### 3.4 JUSTIFICATION

A central element of our methodology is the decision to assess auto-consistency through observable outputs rather than through confidence estimates or internal activations. Probabilities associated with the predicted class quantify only the model's internal belief in its own decision and therefore introduce circularity: the system may appear consistent simply by reiterating the same classification with perturbed confidence, without demonstrating functional dependence on the highlighted tokens. Such tautology undermines the objective of testing auto-consistency.

Moreover, the internal representations of proprietary LLMs are inaccessible in the inference-only setting adopted here, precluding the application of gradient-based or attention-based attribution methods. Under these constraints, the only feasible strategy is to probe consistency through behavioral interventions on the subset $\mathcal{T}(x_i)$ identified as relevant.

Accordingly, our evaluation examines whether removing, isolating, or semantically altering $\mathcal{T}(x_i)$ produces systematic and interpretable changes in the scalar review score $R^{\mathcal{M}}(x)$. This design avoids both the circularity of probability-based measures and the opacity of internal activations, providing a rigorous and reproducible criterion for assessing the auto-consistency of LLM explanations.

## 4 EVALUATION METRICS

The experimental procedure operates at the level of individual sentences, as illustrated in Figure 2. For each input $x_i$, the model first produces a prediction and identifies a subset of influential tokens $\mathcal{T}(x_i)$. Interventions are then applied to construct a modified input $x'_i$, obtained by retaining, removing, or semantically inverting $\mathcal{T}(x_i)$. A second evaluation by the LLM yields an updated score $R^{\mathcal{M}}(x'_i)$, which can be compared against the original score $R^{\mathcal{M}}(x_i)$ to assess the functional role of the highlighted tokens.

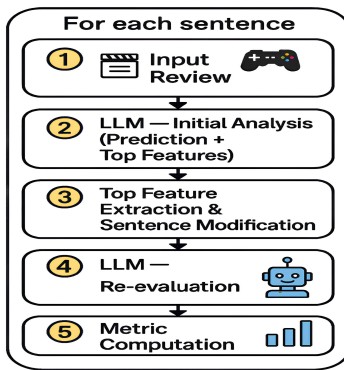

Figure 2: Workflow of interventions and re-evaluations.

While this procedure provides insight into the effect of interventions at the sentence level, drawing conclusions at the scale of an entire dataset requires aggregation. We therefore introduce four complementary measures that summarize consistency across all sentences in a corpus:

- **Mean review difference** ($\overline{\Delta R}$): the average of individual differences $R^{\mathcal{M}}(x_i) - R^{\mathcal{M}}(x'_i)$, reflecting the overall effect of the intervention on the model's evaluation.

- **Standard deviation of differences** ($\sigma_{\Delta R}$): the variability of these differences across sentences.

- **Label-flip proportion** ($\pi_{\text{alt}}$): the fraction of instances where the predicted class changes after intervention.

- **Directional review reduction** ($\pi_{\text{red}}$): the proportion of cases in which the intervention drives the score toward the opposite sentiment polarity. Formally, let $\hat{y}^{\mathcal{M}}(x_i) \in \{\text{positive}, \text{negative}\}$ denote the label predicted for the original sentence. Then:

$$\Delta R_i = \begin{cases} R^{\mathcal{M}}(x_i) - R^{\mathcal{M}}(x_i'), & \text{if } \hat{y}^{\mathcal{M}}(x_i) = \text{positive}, \\ R^{\mathcal{M}}(x_i') - R^{\mathcal{M}}(x_i), & \text{if } \hat{y}^{\mathcal{M}}(x_i) = \text{negative}, \end{cases}$$

with $\pi_{\text{red}}$ computed as:

$$\pi_{\text{red}} = \frac{1}{m} \sum_{i=1}^{m} \mathbb{1}[\Delta R_i > 0].$$

Together, these metrics provide a principled basis for testing the functional alignment between the tokens marked as explanatory and the predictive behavior of the model. Higher values of $\overline{\Delta R}$, $\pi_{\text{alt}}$, or $\pi_{\text{red}}$ indicate stronger dependence on the highlighted subset, while low variability in $\sigma_{\Delta R}$ suggests more consistent explanatory behavior across instances.

## 5 EXPERIMENTAL RESULTS

Table 1 reports the outcomes of the sufficiency, comprehensiveness, and counterfactuality experiments across IMDB and Steam. A consistent trajectory emerges for Gemma3:4B, GPT-4o-mini, and Granite8B: highlighted tokens generally preserve the original decision when isolated, their removal weakens predictions without systematically overturning them, and semantic inversion produces stronger shifts, as expected when the meaning of decisive elements is reversed.

The DeepSeek models deviate from this trajectory. The 1.5B variant shows instability from the outset, with elevated flip rates under sufficiency that persist across the other interventions. The 14B variant, in contrast, maintains greater stability in sufficiency but reacts disproportionately when tokens are removed or inverted, leading to shifts and class changes substantially above those observed in the other models.

Table 1: Summary of auto-consistency metrics on the *IMDB* and *Steam* datasets. Values in **bold** correspond to specific results explicitly discussed in section 6.

| | IMDB | | | | Steam | | | |
|---|---|---|---|---|---|---|---|---|
| **Model** | $\overline{\Delta R}$ | $\sigma_{\Delta R}$ | $\pi_{\text{alt}}$ | $\pi_{\text{red}}$ | $\overline{\Delta R}$ | $\sigma_{\Delta R}$ | $\pi_{\text{alt}}$ | $\pi_{\text{red}}$ |
| **Sufficiency** | | | | | | | | |
| Gemma3:4B | 0.02 | 1.48 | 3.8 | 14.2 | 0.27 | 1.84 | 6.9 | 16.4 |
| GPT-4o-mini | -0.28 | 0.87 | **0.9** | 10.5 | -0.09 | 1.33 | **3.5** | 13.3 |
| Granite8B | -0.41 | 1.42 | 3.8 | 9.9 | 0.13 | 1.89 | 6.7 | 21.0 |
| DeepSeek-1.5B | 0.66 | 2.84 | **29.1** | 43.6 | 0.56 | 2.91 | **30.9** | 42.9 |
| DeepSeek-14B | -0.26 | 1.51 | 13.7 | 21.0 | -0.02 | 1.75 | 10.6 | 22.3 |
| **Comprehensiveness** | | | | | | | | |
| Gemma3:4B | 0.91 | 1.96 | 15.7 | 32.1 | 1.74 | 2.82 | 24.5 | 44.3 |
| GPT-4o-mini | 1.21 | 1.70 | **19.9** | 53.9 | 1.80 | 2.71 | **28.5** | 50.0 |
| Granite8B | 0.54 | 1.21 | 6.1 | 35.1 | 1.54 | 2.62 | 20.3 | 48.0 |
| DeepSeek-1.5B | 0.55 | 2.66 | **39.0** | 38.9 | 0.89 | 2.88 | **39.4** | 44.3 |
| DeepSeek-14B | 0.62 | 1.42 | **31.5** | 44.2 | 1.68 | 2.76 | **45.5** | 45.5 |
| **Counterfactuality** | | | | | | | | |
| Gemma3:4B | 2.04 | 2.76 | 32.4 | 49.7 | 2.71 | 3.25 | 41.5 | 50.5 |
| GPT-4o-mini | 2.43 | 2.41 | **43.6** | **65.9** | 2.28 | 2.72 | **38.3** | **57.0** |
| Granite8B | 1.50 | 2.20 | 24.3 | 53.5 | 2.36 | 3.14 | 33.9 | 54.4 |
| DeepSeek-1.5B | 0.89 | 2.51 | **45.4** | **45.9** | **1.16** | 2.87 | **44.9** | **44.9** |
| DeepSeek-14B | 1.90 | 2.20 | **49.4** | 60.3 | **2.65** | 3.08 | **49.4** | 49.4 |

Taken together, the results distinguish two profiles. Gemma3, GPT-4o-mini, and Granite8B follow the expected progression of increasing sensitivity across interventions, whereas the DeepSeek models diverge—one by failing to preserve stability under sufficiency, the other by exhibiting excessive volatility under stronger perturbations.

## 6 DISCUSSION

The joint examination of sufficiency, comprehensiveness, and counterfactuality highlights systematic differences in how models maintain alignment between the tokens they mark as explanatory and

their predictive behavior. For Gemma3:4B, GPT-4o-mini, and Granite8B, the results follow a progression consistent with expectations. Under sufficiency, label-flip rates remain below 10% in both datasets, indicating that the highlighted tokens alone generally sustain the original prediction. Comprehensiveness introduces moderate instability, with class-change rates rising to around 20–28% (e.g., 19.9% for GPT-4o-mini on IMDB and 28.5% on Steam), showing that the removal of these tokens weakens predictions but does not uniformly overturn them. Counterfactuality then produces stronger disruptions, with flip rates between 24–43%, confirming that semantic inversion systematically impacts predictions while still remaining within interpretable ranges. This trajectory suggests that these models distribute importance between the highlighted terms and broader contextual cues, yielding explanations that, while not exhaustive, remain functionally meaningful.

By contrast, the open-source DeepSeek models reveal systematic departures from this pattern. The 1.5B variant exhibits structural inconsistency: in sufficiency, nearly one-third of sentences change class (29.1% on IMDB and 30.9% on Steam), showing that the highlighted tokens are not enough to preserve the original prediction. Yet in comprehensiveness, their removal produces equally high class-change rates (39–40%), implying that the same tokens are indispensable. This contradiction is compounded in counterfactuality, where high flip rates (around 45%) coincide with relatively small score differences, indicating that minimal semantic edits frequently overturn predictions without systematically steering them toward the opposite polarity.

The 14B variant reflects a distinct but equally problematic behavior. While it maintains moderate stability in sufficiency (13.7% on IMDB and 10.6% on Steam), comprehensiveness introduces sharp increases in class-change rates (31.5–45.5%), and counterfactual interventions amplify this effect, with flip rates approaching 50% and large shifts in review scores (up to 3.77 on Steam). In this case, predictions hinge disproportionately on the highlighted tokens, collapsing under perturbations that other models absorb with greater stability.

Taken together, these findings show that auto-consistency in LLM explanations is not guaranteed by either model scale or openness. GPT-4o-mini displays the clearest alignment with the theoretical expectations of the metrics, while Gemma3:4B and Granite8B achieve partial but coherent consistency. The DeepSeek models, however, illustrate two qualitatively distinct modes of failure: contradiction in the smaller variant and over-reliance in the larger. These results emphasize the need for interpretability research to move beyond textual plausibility and to adopt evaluation strategies that directly test whether the tokens marked as explanatory play a functionally stable role in the decision-making process.

## 6.1 LIMITATIONS AND FUTURE WORK

This study has several limitations that constrain the scope of its conclusions. First, the evaluation was limited to two sentiment-analysis datasets (IMDB and Steam), which provides a controlled setting but restricts generalization to domains with different linguistic and semantic properties. Second, the proposed metrics focus on token-level interventions and do not capture higher-level structures such as syntax, discourse, or compositional semantics, which may also affect stability. Third, the extraction of explanatory tokens itself may introduce inconsistencies, particularly in smaller models, complicating comparisons across metrics.

Future work should extend the analysis to a broader range of datasets and model families, refine counterfactual interventions with richer semantic resources, and combine these behavioral metrics with human-centered evaluations, thereby assessing not only internal coherence but also the external utility of the explanations. Despite these limitations, the results presented here provide a rigorous and reproducible basis for evaluating auto-consistency in LLM explanations, highlighting systematic differences across architectures and parameter scales. This contribution advances the methodological toolkit for probing the functional reliability of LLM explanations and offers empirical evidence that can guide future developments in interpretability research.

## ETHICS STATEMENT

The authors state they read the ICLR Code of Ethics and adhere to it.

REPRODUCIBILITY STATEMENT

Ensuring reproducibility is central to the reliability of empirical research in interpretability. All experiments were conducted using Python 3.13 on a Linux machine (Ubuntu 24.04.3 LTS) with an AMD Ryzen 7 5700X CPU, 32 GB of RAM, and an NVIDIA RTX 3060 Ti GPU (8 GB).

The complete codebase is available at [the link will be shared after the reviewing process], including the exact prompts (in the prompts directory) and datasets (in the data directory). We employed two sentiment-analysis benchmarks: the IMDB Large Movie Review Dataset (Maas et al., 2011), accessible via HuggingFace,[6] and the Steam Review Dataset (Pandey & Joshi, 2022), available on Kaggle.[7]

Models included both open-source LLMs (gemma3, granite-8b, and the deepseek family with 1.5B and 14B parameters, executed locally via Ollama) and the closed-source GPT-4o-mini, accessed through the OpenAI API.[8]

This setup, together with the released resources, ensures that our experiments can be replicated and extended to alternative datasets, models, or prompting strategies.

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

APPENDIX

## A    PROMPT USED

To ensure comparability across distinct types of review data, we employ a parameterized prompt template. A domain variable, denoted as **DOMAIN**, is instantiated as either movie or game, corresponding respectively to film or game reviews. This parametrization enables the reuse of a single prompt structure with minimal modification, while preserving uniformity in task specification and response formatting.

The prompt further incorporates a placeholder for few-shot examples. These examples are not merely illustrative but are embedded directly within the prompt to condition the model's behavior. In this way, the examples serve as explicit demonstrations of the expected output structure and reinforce task alignment.

For clarity, we present (i) the general prompt template, which includes placeholders for both the domain specification and the examples, and (ii) the domain-specific instantiations of these placeholders, which operationalize the template for the two datasets considered.

---

**Prompt Template**

You are a sentiment analysis assistant working as a classifier for **{{DOMAIN}}** reviews.
Your tasks are:
1. Analyze the input sentence and decide "positive" or "negative".
2. Return "sentiment" as "positive" or "negative".
3. Provide a **review score** "review" from 1 to 10. Higher means stronger approval.
4. List in "influential_terms" the words or phrases directly extracted from the sentence that were important for the prediction.
5. Include the original review in "input_sentence".
**STRICT OUTPUT RULES**
• Output must be valid JSON only.
• Do not include Markdown fences or explanations.
• Use straight quotes " and escape inner quotes with \".
• Output must start with { and end with }.
• Output must contain exactly these fields, in any order: sentiment, review, influential_terms, input_sentence.
**Examples:**
{{EXAMPLES}}

---

**Examples**

**Case {{DOMAIN}} = movie**

```
{
  "sentiment": "positive",
  "review": 10,
  "influential_terms": [
    "greatest episode",
    "What a series",
    "speechless"
  ],
  "input_sentence":
    "This is the greatest episode I've ever watched.
     What a series! I'm speechless."
}

{
  "sentiment": "negative",
  "review": 5,
  "influential_terms": [
    "not awful",
    "not great",
    "not sure why"
```

```
    ],
    "input_sentence":
      "It is not awful, not great either...
       I am honestly not sure why some people like it so much."
}
```

**Case {{DOMAIN}} = game**

```
{
    "sentiment": "positive",
    "review": 10,
    "influential_terms": [
      "incredible gameplay",
      "tight controls",
      "runs flawlessly"
    ],
    "input_sentence":
      "One of the best games I have ever played.
       Tight controls and it runs flawlessly
       even on high settings."
}

{
    "sentiment": "negative",
    "review": 2,
    "influential_terms": [
      "pay-to-win",
      "constant crashes",
      "unbalanced multiplayer"
    ],
    "input_sentence":
      "Pay-to-win mechanics, constant crashes,
       and unbalanced multiplayer ruined
       the experience for me."
}
```

## B  USE OF LARGE LANGUAGE MODELS (LLMS)

LLMs were employed exclusively to support the preparation of this manuscript in non-substantive ways. Specifically, they were used to assist in the revision of textual coherence, refinement of academic writing style, and formatting consistency. In addition, LLMs were applied to generate preliminary drafts of figures and tables, which were subsequently reviewed and finalized by the authors. At no stage were LLMs involved in the design of experiments, execution of analyses, or interpretation of results. All methodological decisions, experimental procedures, and substantive contributions remain the responsibility of the authors. The use of LLMs was thus limited to editorial assistance and visualization support, ensuring that the scientific content of the article is fully author-driven.

