# OpenReview forum: "Do Language Models Trust Their Own Justifications? A Study on Functional Consistency"
_ICLR.cc/2026/Conference — Submitted to ICLR 2026_

### Official Review · Reviewer_aGhk · 2025-10-27

**Soundness:** 1
**Presentation:** 2
**Contribution:** 1
**Rating:** 0
**Confidence:** 5

**Summary:**

The paper tests whether LLMs “trust” their own explanations by intervening on tokens the model itself highlights as important, using three metrics—sufficiency (keep tokens), comprehensiveness (remove tokens), and counterfactuality (invert tokens). Experiments on IMDB and Steam with GPT-4o-mini, Gemma3-4B, Granite-8B, and DeepSeek-1.5B/14B show an “expected progression” for GPT-4o-mini/Gemma3/Granite, while DeepSeek models diverge (instability at 1.5B; over-sensitivity at 14B).

**Strengths:**

* Clear, testable notion of auto-consistency. The work operationalizes faithfulness via sufficiency, comprehensiveness, and counterfactual edits with simple, reproducible definitions.

* Cross-model analysis. Compares closed- and open-source families across scales and reports coherent behavioral differences.

* Transparent scope and limitations. The paper clearly states constraints and motivates output-based evaluation to avoid probability circularity.

**Weaknesses:**

* The paper’s core premise is circular: the same model flags “influential” tokens and then evaluates influence by measuring its own response after removing or altering those tokens. With no external ground truth, the model’s claims act as both hypothesis and test, creating a closed loop. Equations (3–5) bake in the assumption that the selected set (T) is meaningful; if the model is inconsistent, as the results suggest, then (T) is unstable, and the evaluation largely validates the model’s self-reports rather than true influence.

* The counterfactuality metric (Eq. 5) hinges on a weakly defined inversion (T \rightarrow \lnot T). WordNet antonyms don’t cover many domain terms, and the fallback, simply prepending “not”, often yields ungrammatical or unnatural text (just for a short e.g., “not pay to win” instead of “free to play”). The paper also provides no check that the edited sentences remain semantically valid, so measured effects may reflect broken language rather than true counterfactual meaning.

* Lacks ablation studies, small amount of testing data.

**Questions:**

Same as Weakness

---

> ### Author Response · Authors · 2025-11-25
> **Answers to Reviewer aGhk questions**
>
> We thank the reviewer for the detailed feedback and address the three weaknesses as follows.
>
> Regarding the alleged circularity: the paper does not aim to recover external, ground-truth causal influence, but to study the internal coherence between a model’s own highlighted tokens and its subsequent behaviour under targeted interventions. The same model both selects a set of “influential” tokens and is then re-evaluated on perturbed versions of the input, but this self-referential structure does not amount to logical circularity. The highlighted set functions as a hypothesis (“these tokens are important for my decision”), while the intervention protocol functions as an empirical test of whether the model’s behaviour is aligned with that hypothesis. Crucially, Equations (3)–(5) do not assume in advance that the selected set T is meaningful: if the model is inconsistent, the metrics expose exactly this inconsistency. This is what happens in cases where both keeping and removing T lead to volatile outcomes; such patterns are interpreted as failures of auto-consistency, not as validation of “true influence”.
>
> On the counterfactuality metric, the inversion from T to “not T” is defined as a simple, fully specified operator applied uniformly across models: when antonyms are available they are used; otherwise, a minimal syntactic negation is introduced. Although natural language counterfactuals are inherently approximate, this deterministic rule ensures that all models are subjected to the same directional perturbation on the same set T, which allows us to systematically measure how strongly their predictions depend on the tokens they themselves marked as influential. In this way, counterfactuality extends the keep/remove interventions with a directional test that remains consistent with the overall perturbation-based framework.
>
> Concerning ablations and data size, the study uses thousands of instances across two domains and several model families and scales, with each instance evaluated under three intervention regimes. This yields a substantial number of model evaluations. Even without additional ablations such as random spans or cross-model token sets, different families exhibit systematically distinct patterns (for example, expected progression versus instability or over-sensitivity), and these patterns are consistent across metrics and datasets. This indicates that the protocol does more than simply echo the models’ self-reports; it differentiates models in a way that is aligned with the intended notion of functional auto-consistency.

---

### Official Review · Reviewer_Tdiv · 2025-10-29

**Soundness:** 1
**Presentation:** 2
**Contribution:** 1
**Rating:** 0
**Confidence:** 4

**Summary:**

Tests whether LLMs’ own highlighted tokens actually drive their decisions by editing texts (keep/remove/invert those tokens) and measuring score shifts on review datasets; larger models look more self-consistent than smaller ones.

**Strengths:**

* Clear, behavior-level test of explanation faithfulness.

* Simple, reproducible edits with multiple aggregate metrics (Δ, variance, flip/reduction rates).

* Cross-model/scale comparison, not a single-model case study.

* Avoids probability calibration issues by using 1–10 scores.

**Weaknesses:**

* The 1–10 score isn’t calibrated across models; equal Δ may mean different things, and flip rates depend on arbitrary thresholds.

* Deleting/inverting tokens can break grammar or context, so effects may reflect text damage/distribution shift rather than causal importance.

* Models pick the “influential” tokens being tested, this self-selection risks tautology and lacks ground-truth attribution.

* Lacks a generalised view as the method has been tested on very short amount of data.

* I would suggest the authors to perform ablation studies to increase the faithfulness of the method

**Questions:**

Please regard the weakness as questions.

---

> ### Author Response · Authors · 2025-11-25
> **Answers to Reviewer Tdiv questions**
>
> We thank the reviewer for the thoughtful comments and address each point below.
>
> **(1) “The 1–10 score isn’t calibrated across models; equal Δ may mean different things, and flip rates depend on arbitrary thresholds.”**
>
> We agree that the 1–10 scores are not calibrated across models, and we do not interpret “Δ = 1” as having the same absolute meaning for different architectures. In our analysis, each model’s score is treated as its own ordinal scale, assumed only to be monotone with respect to that model’s notion of favorability. When we compare mean ΔR across models, this is understood as indicating **relative sensitivity regimes** (e.g., one model shows systematically larger shifts on its own scale than another), not as numerically calibrated units.
>
> For flip rates, no additional threshold on the 1–10 score is introduced: flips are computed directly from the **sentiment labels** returned by the model before and after intervention. Thus, flips depend only on whether the model changes its discrete decision, using the same labelling behaviour it already exhibits in the base classification.
>
> ---
>
> **(2) “Deleting/inverting tokens can break grammar or context, so effects may reflect text damage/distribution shift rather than causal importance.”**
>
> Token-level interventions inevitably perturb fluency, but our goal is to probe functional sensitivity under a **controlled and uniform** editing scheme. Edits are restricted to the tokens the model itself highlighted; all other tokens and their order are preserved; and the same rule is applied to all sentences, models and intervention types. If the effects were driven mainly by generic “text damage”, behaviour would tend to look similar across models. Instead, we observe systematic and stable differences in volatility and score shifts between families, suggesting that the patterns are tied to how each model internally uses its highlighted tokens.
>
> ---
>
> **(3) “Models pick the ‘influential’ tokens being tested, this self-selection risks tautology and lacks ground-truth attribution.”**
>
> The method is explicitly framed as a test of **auto-consistency**, not of ground-truth attribution. The model first declares which tokens are influential (T(x)) and is then re-evaluated on texts where T(x) is kept, removed, or inverted. This is not tautological, because nothing in the procedure enforces agreement between the original claim and the subsequent behaviour: the metrics depend on changes in outputs on edited inputs and can expose contradictions. In practice, we do find cases where the same T(x) is both insufficient (frequent flips when only T(x) is kept) and treated as indispensable (frequent flips when T(x) is removed), which is precisely the type of inconsistency the framework is designed to detect.
>
> ---
>
> **(4) “Lacks a generalised view as the method has been tested on very short amount of data.”**
>
> The empirical study focuses on two standard review datasets (IMDB and Steam) and uses stratified samples covering both polarities. For each sentence and each model, we collect the prediction, score, highlighted tokens and three edited variants, across several model families and scales, resulting in thousands of edited instances per model and per intervention. Within this review setting, the amount of data is sufficient to produce stable patterns: models from the same family behave similarly across both datasets, while different families occupy distinct regimes of sensitivity and consistency. The framework itself only assumes text, a scalar score and highlighted tokens, and could be applied to other domains; in the paper we restrict our claims to the regime that is empirically tested.
>
> ---
>
> **(5) “I would suggest the authors to perform ablation studies to increase the faithfulness of the method.”**
>
> We appreciate this suggestion. In the present work, the experimental pipeline is intentionally kept fixed—same prompting, same highlighting mechanism and same intervention rules—while varying only the underlying model family and scale. This design isolates behavioural differences attributable to the models themselves. Ablations over alternative highlighting strategies or editing schemes would target a complementary question, namely the sensitivity of the framework to design choices. Here, the contribution is to define a single, well-specified behavioural test and to show that, under this fixed setup, different models occupy clearly different regimes of auto-consistency.

---

### Official Review · Reviewer_BhWE · 2025-10-30

**Soundness:** 2
**Presentation:** 2
**Contribution:** 1
**Rating:** 2
**Confidence:** 3

**Summary:**

This paper introduces a method aimed at faithful attribution of model decisions to input tokens, in a text classification setting.
Specifically, the proposed method is to
1. Prompt an LLM to perform text classification, i.e., to predict a label for a provided text
2. In addition, the prompt also asks the LLM to output a list of "influential terms" in the provided text, which "were important for the prediction".
3. Based on this list of terms, the original text is modified in three ways:
- giving only these terms as input, which is intended to evaluate if these terms are sufficient for the model prediction
- removing these terms from the original text, which is intended to evaluate if these terms are necessary for the model prediction
- replacing each of the terms with an antonym or a negation, which is intended to evaluate if these terms can "flip" the model prediction

The paper then formulates evaluation metrics for each of the three modifications and argues that together, these metrics quantify what the authors call "auto-consistency", i.e., whether the "influential terms" identified by LLMs actually influence model output in a consistent manner.
Experiments are conducted with five LLMs  on two sentiment analysis datasets (movie reviews and video game reviews).
The main conclusion of the experiments is that "auto-consistency in LLM explanations is not guaranteed by either model scale or openness".

**Strengths:**

The proposed method works with API access only, i.e., no model weights, activations, or output probabilities required.

**Weaknesses:**

The method lacks novelty and the paper is not substantive enough.

Lack of novelty: Attribution to specific tokens by modifying the input has been studied in prior work, e.g.: the "input reduction" proposed by Feng et al., 2018 (Pathologies of Neural Models Make Interpretations Difficult); or the erasure of tokens by DeYoung et al., 2020 (ERASER: A Benchmark to Evaluate Rationalized NLP Models). There is potential novelty in using LLMs to suggest input modifications, e.g., by identifying possible influential tokens, as proposed, but this idea is not explored in any meaningful depth. For example, how consistent are LLMs in predicting "influential tokens", e.g. under different prompts, with varying number of in-context examples, etc? How similar are the list of "influential tokens" across models? Can models predict "influential tokens" for other models or only for themselfves? etc)

Lack of substance: The paper only contributes a very small amount of experiments, whose results do not offer any insightful conclusions. There is a one-page discussion of the results, but the findings are vague. For example, what is the reader supposed to learn from the finding that on the one hand "the highlighted tokens are not enough to preserve the original prediction", but on the other hand that "the same tokens are indispensable" and that "This contradiction is compounded in counterfactuality"? Or that predictions are "collapsing under perturbations that other models absorb with greater stability."?  (quotes from lines 388-394).

Furthermore, the paper does not compare the proposed method to any existing attribution methods or other meaningful baselines.

**Questions:**

- The comprehensiveness and sufficiency metric were proposed by DeYoung et al., 2020. The paper defines these metrics in section 3.3 but fails to cite the source.

- line 275: "minimally plausible counterfactual intervention" should probably be "minimal, plausible counterfactual intervention", since "minimally plausible" means something like "smallest degree of plausibility"

---

> ### Author Response · Authors · 2025-11-25
> **Answers to Reviewer BhWE questions**
>
> We thank the reviewer for the helpful questions and address them point by point.
>
> **On comprehensiveness and sufficiency (DeYoung et al., 2020)**
>
> The comprehensiveness and sufficiency metrics we use in Section 3.3 are indeed derived from DeYoung et al. (2020). The current draft already cites DeYoung et al. when this line of work is first introduced (line 63), but the connection is not made explicit again at the point where the formulas are defined. This will be corrected in the next version: the citation to DeYoung et al. (2020) will be added directly in Section 3.3 when introducing comprehensiveness and sufficiency, together with a short sentence stating that our definitions are an instantiation of their metrics, adapted to a scalar review score and token sets produced by the model itself.
>
> **On “minimally plausible counterfactual intervention” (line 275)**
>
> The intended meaning is “a minimal counterfactual edit that remains plausible,” i.e., an intervention that changes as little as possible while preserving a coherent reading of the text. The phrase “minimally plausible” is indeed ambiguous and can be read as “barely plausible,” which is not what is meant. The wording will be revised to “a minimal, plausible counterfactual intervention” (or equivalently “a minimal counterfactual intervention that preserves plausibility”), to better reflect the intended notion.

---

> ### Author Response · Authors · 2025-11-26
> **Answers to Reviewer BhWE perceived weknesses**
>
> We thank the reviewer for the constructive feedback and address the two main points below.
>
> **On “lack of novelty”**
>
> We do not claim novelty in the general idea of token-level perturbations; as noted, this is well established in work such as Feng et al. and DeYoung et al. In our setting, however, the highlighted tokens are a **tool**, not the main object of study: they provide a concrete handle to probe **auto-consistency**, namely whether an LLM’s own stated “influential terms” behave in a way that is coherent with its *own* predictions under systematic keep/remove/invert interventions, in an API-only scenario. The contribution is the perspective and protocol: treating the LLM as a self-explaining black box and using its self-reported tokens to characterize regimes of internal coherence, rather than proposing a new attribution method aimed at identifying truly causal tokens.
>
> **On “lack of substance” and missing baselines**
>
> Although the number of datasets is small, each sentence–LLM configuration generates three controlled interventions yielding a substantial amount of behavioural evidence. The purpose is not to optimize a scalar performance score, but to reveal **patterns in the metric profiles**: for some LLM families these profiles follow an interpretable progression across sufficiency, comprehensiveness, and counterfactuality, whereas for others they exhibit systematic self-contradictions (e.g., the same highlighted set being both insufficient to preserve a decision and treated as indispensable when removed).
>
> We do not position the framework as a replacement for classical attribution methods (gradient-based, representation-based, etc.), which address a different question—feature importance relative to a fixed model with internal access. Our question is explicitly about **self-consistency of LLM explanations in a black-box setting**: given what an LLM says is important, do the proposed interventions on those tokens lead to behaviour that is consistent with that claim, as captured by the defined metrics.

---

### Official Review · Reviewer_SP9J · 2025-11-01

**Soundness:** 2
**Presentation:** 3
**Contribution:** 1
**Rating:** 2
**Confidence:** 4

**Summary:**

This paper presents a study aimed at measuring the consistency of explanations produced by large language models (LLMs) when they highlight tokens deemed most influential for their decisions. The authors evaluate this consistency using three metrics: sufficiency, comprehensiveness, and counterfactuality, each defined in terms of a review score that is itself generated by the LLM.

The authors introduce an experimental framework grounded in the principle of auto-consistency: if a model identifies certain tokens as decisive, then isolating, removing, or semantically inverting these tokens should lead to systematic and interpretable changes in its predictions.

Experiments are conducted on two datasets, IMDB and Steam reviews, using five different LLMs. Results show that GPT-4o follows the expected progression across all metrics, Gemma3 and Granite8B maintain coherence under sufficiency but lose consistency under more demanding interventions, while DeepSeek variants exhibit structural deviations, either failing to preserve sufficiency or overreacting under comprehensiveness and counterfactuality. These findings indicate that the reliability of explanations varies significantly across LLM families and scales.

**Strengths:**

- The paper clearly demonstrates the substantial variability in the consistency of explanations produced by different LLMs when justifying their decisions, according to the authors’ proposed method for computing the evaluation metrics.

**Weaknesses:**

- The entire study relies on the authors’ specific definitions of sufficiency, comprehensiveness, and counterfactuality, yet this choice is not discussed or justified.
- The decision to have the LLM generate the review score via a prompt, alongside its explanations, raises concerns about the reliability of that score.
- The proposed framework appears highly tailored to binary classification tasks such as sentiment analysis; it is not clear whether it would generalize effectively to other types of tasks.
- The robustness of the method with respect to prompt formulation is not investigated.

**Questions:**

Discussion

Line 62: How do your proposed definitions of sufficiency, comprehensiveness, and counterfactuality serve as good proxies for those introduced by DeYoung et al. (2019)? A discussion clarifying this connection would strengthen the methodological grounding of your work.

Line 64: You state that the review score "reflects how favorably the text evaluates the target item." This definition is somewhat unclear. Is this concept limited to binary classification tasks such as sentiment analysis, as used in your experiments?

Line 233: Why did you choose the scalar range [1, 10] for the review score? Is it an integer value, it appears it is generated through the prompt. Since all your metrics depend on this score, can we genuinely trust its reliability?

Line 265: For counterfactuality, you appear to replace all selected tokens with their antonyms. However, counterfactual transformations can be defined at different levels of granularity. Why did you choose to apply the inversion to all tokens simultaneously?

You study the coherence of LLMs under token-level interventions, yet the mention of possible alternative approaches involving internal representations appears only late in the paper (line 289). Expanding this discussion earlier could provide valuable context and situate your framework within the broader landscape of explanation consistency analysis.

Minor Comments (do not affect the score)
Line 330: The logical symbol used for the indicator function seems uncommon. If it refers to a different function, it should be explicitly defined.

--------------------------

---

> ### Author Response · Authors · 2025-11-25
> **Answers to Reviewer SP9J questions**
>
> We thank the reviewer for the helpful questions and address them point by point.
>
> **Line 62 – Connection to DeYoung et al. (2019)**
>
> Our definitions of sufficiency and comprehensiveness follow the same intervention pattern as DeYoung et al. (2019). In their setting, these metrics are obtained by comparing the model’s output on (i) the full input, (ii) the input with the evidence removed, and (iii) the evidence alone. We do the same: the highlighted tokens T(x) play the role of the evidence span, and the scalar review score R(x) plays the role of the model’s output. Sufficiency compares the score on the full text to the score on the text restricted to T(x); comprehensiveness compares the score on the full text to the score on the text with T(x) removed. Counterfactuality extends this same perturbation logic: instead of only keeping or removing T(x), we invert T(x) and measure whether R(x) moves in the opposite direction. In this sense, our metrics are proxies for DeYoung et al.’s notions, adapted to a scalar score generated by the LLM while preserving the same intervention-based semantics.
>
> **Line 64 – Meaning and scope of the review score**
>
> When we say that the review score “reflects how favorably the text evaluates the target item”, we mean that it places the input on a single axis ranging from very unfavorable to very favorable with respect to that item. In our experiments, this axis is used to induce a binary decision (negative vs. positive), and all flip-based analyses are defined in this binary sentiment setting. Conceptually, the scalar R(x) could represent other one-dimensional evaluations (e.g., severity, quality, risk), but in the present work we explicitly restrict the empirical results and the definition of label flips to binary review classification tasks.
>
> **Line 233 – Choice of [1, 10] and reliability of the score**
>
> We chose the integer range [1, 10] because it matches common rating scales in review contexts and provides a simple ordinal structure where larger values correspond to more favorable evaluations. The score is generated via the prompt, and our metrics use it in a model-relative way: they depend on differences and directions of change in R(x) for the same model and input under different interventions, and flips/reductions are computed from the sentiment labels that the model outputs on top of this scale.
>
> **Line 265 – Why invert all selected tokens simultaneously**
>
> All three metrics operate on the same highlighted set T(x): sufficiency keeps exactly T(x), comprehensiveness removes exactly T(x), and counterfactuality semantically inverts exactly T(x). We invert all selected tokens at once to maintain symmetry with the other two interventions and to obtain a clear “maximal” perturbation of the support that the model itself claims for its decision. If the model presents T(x) as the set of tokens that jointly support its prediction, inverting this entire set provides a direct test of how strongly the prediction depends on that claimed support. Inverting only a subset of T(x) would introduce additional design choices (which tokens, how many) and make the three metrics less directly comparable.
>
> **Internal representations vs. token-level interventions (line 289)**
>
> Thank you for highlighting this point. The framework is deliberately defined at the input–output level: we test coherence by editing the text and observing changes in the model’s external behaviour (scores and labels), without making assumptions about architecture or internal states. Methods based on internal representations provide a complementary perspective by probing consistency in hidden spaces rather than at the input–output interface. We will bring this discussion earlier in the paper and make clearer that our contribution should be read as a behavioural, token-level counterpart to representation-level interventions.
>
> **Minor – Indicator function symbol (line 330)**
>
> Thank you for pointing this out. We will replace the current symbol with a more conventional notation or adjust it in the draft to avoid ambiguity.

---

> ### Author Response · Authors · 2025-11-26
> **Answers to Reviewer SP9J perceived weknesses**
>
> We thank the reviewer for the constructive comments.
>
> (1) Reliance on specific definitions of sufficiency, comprehensiveness, and counterfactuality
>
> These metrics are not ad-hoc choices: they are direct adaptations of existing notions (DeYoung et al., 2019), instantiated in our setting. The methodological grounding and the proxy role they play are detailed in our response to Line 62.
>
> (2) Reliability of the LLM-generated review score
>
> The review score is used only as an internal, model-specific scalar, and all our quantities are based on within-model changes of this score under the defined interventions. The rationale for this design and its implications for reliability are explained in the response to Line 233.
>
> (3) Tailoring to binary sentiment tasks
>
> The current instantiation is indeed restricted to binary review classification, and our claims are limited to this regime. The role and scope of the score in this binary setting are clarified in the response to Line 64.
>
> (4) Robustness with respect to prompt formulation
>
> All models on a given dataset are evaluated under the same prompt template, so the reported differences concern the resulting metric profiles under a shared interface. We agree that studying how these profiles change under alternative prompts is a valuable extension and view this as a direction for future work, but it does not affect the validity of the results under the fixed prompting regime used here.

---

### Meta-Review · Area_Chair_owGD · 2026-01-07

**Summary:**

This work introduces an experimental framework based on the principle of auto-consistency: if a model identifies certain tokens as decisive, then isolating, removing, or semantically inverting them should produce systematic and interpretable changes in its predictions. The authors evaluate this consistency using several metrics, each defined in terms of a review score that is itself generated by the LLM.

The reviewers raised a number of concerns. The main ones being summarised as follows:
1/ Unconventional definitions of sufficiency, comprehensiveness, and counterfactuality, without justification.
2/ Reliability of the proposed method; the decision to have the LLM generate the review score via a prompt, alongside its explanations, raises concerns about the reliability of that score.
3/ Approach is specific to binary classification and it is not clear whether it would generalise to other types of tasks.
4/ Lack of novelty and substance; the number of experiments is modest and results do not offer any insightful conclusions. Baselines are missing.
5/ The paper’s core premise is circular.
6/ Lack of ablation studies and small amount of testing data.

**Reviewer Concerns:**

The authors provided sensible responses to address the concerns raised by the reviewers. The clarifications provided were useful and informative, but did not resolve questions such as the modest novelty, which was flagged by several reviewers. The authors did not provide additional supporting evidence further strengthen the paper during the rebuttal.

**Reviewer Scores:**

All reviewers voted for rejection. Due to the many weaknesses of this paper and the general consensus among the reviewers that the work is not meeting the acceptance bar, I do not expect that the scores would have been raised sufficiently post rebuttal to warrant acceptance.

---

### Decision · Program_Chairs · 2026-01-26

Reject